# Impact of the COVID-19 Pandemic on the Incidence and Characteristics of Patients with Psychiatric Illnesses Visiting Emergency Departments in Korea

**DOI:** 10.3390/jcm11030488

**Published:** 2022-01-18

**Authors:** Sun Young Lee, Young Sun Ro, Joo Jeong, Sang Do Shin, Sungwoo Moon

**Affiliations:** 1Public Healthcare Center, Seoul National University Hospital, Seoul 03080, Korea; sy2376@gmail.com; 2Department of Emergency Medicine, Seoul National University Hospital, Seoul 03080, Korea; sdshin@snu.ac.kr; 3National Emergency Medical Center, National Medical Center, Seoul 04564, Korea; sungwoo.moon89@gmail.com; 4Department of Emergency Medicine, Seoul National University Bundang Hospital, Seongnam 13620, Korea; yukijeje@gmail.com; 5Department of Emergency Medicine, Korea University Ansan Hospital, Ansan 15355, Korea

**Keywords:** COVID-19, mental health, suicide, emergency

## Abstract

The global coronavirus disease 2019 (COVID-19) pandemic has had wide-ranging effects on the mental health of the public. This study aimed to investigate the impact of the COVID-19 pandemic on the characteristics of psychiatric patients who visited emergency departments (ED) during this time. A cross-sectional study was conducted including patients visiting 402 nationwide EDs from 27 January 2020 to 29 June 2020 (22 weeks; during-COVID) and the corresponding period in 2019 (28 January 2019 to 30 June 2019, 22 weeks; before-COVID) to control for seasonal influences. Among the 6,210,613 patients who visited the ED, 88,520 (2.5%) patients who visited before the pandemic and 73,281 (2.7%) patients who visited during the pandemic had some kind of psychiatric illness. The incidence rates of psychiatric self-harm increased from 0.54 before the pandemic to 0.56 during the pandemic per 1,000,000 person-days (*p* = 0.04). Age- and sex-standardized rates of psychiatric illnesses per 100,000 ED visits increased during the pandemic (rate differences (95% CIs); 45.7 (20.1–71.4) for all psychiatric disorders and 42.2 (36.2–48.3) for psychiatric self-harm). The incidence of psychiatric self-harm and the proportion of psychiatric patients visiting EDs increased during the COVID-19 pandemic.

## 1. Introduction

The global coronavirus disease 2019 (COVID-19) pandemic has had a huge impact on peoples’ daily lives [1]. Several countries implemented nationwide lockdowns and issued social distancing guidelines to reduce person-to-person contact and prevent the spread of the virus [2]. This has led to a reduction in the frequency of social activities and changes in the pattern of use of healthcare services. Several studies have reported a decrease in overall emergency department (ED) visits during lockdown periods and variances in the use of healthcare services depending on the illness and symptoms [3,4]. Motor vehicle incidents have decreased, as has social activity, whereas there has been an increase in the number of ED visits with the presentation of respiratory symptoms [3].

The pandemic has had wide-ranging effects on the mental health of the public [5,6,7,8]. Patients infected during the pandemic have experienced mental stress, such as stigma, not only at the time of infection, but also up to 12 months after the initial infection [9,10]. As the number of people in quarantine have increased during the pandemic, isolated people tend to experienced various negative psychological effects from quarantine, including posttraumatic stress disorder, confusion, and anger [11]. People who are not infected or who have not had direct contact with an infected person have also been psychologically affected by the fear of infection and social isolation due to social distancing [8,12,13,14,15]. The pandemic has adversely affected the mental health of everyone who has lived through it, by making people feel anxious, depressed, and stressed [8,16].

Various risk factors can lead to worsening mental health during a pandemic. In particular, pre-existing mental disorders are a major risk factor for deteriorating mental health during the pandemic [17,18]. For vulnerable populations, mental health may worsen not only due to the pandemic itself, but also because of isolation due to social distancing policies, stress caused by the closure of schools and workplaces, and a decrease in income [13,18,19,20]. These risk factors could promote the sudden onset of symptoms of mental illnesses [5,20]. However, few studies have been conducted on the trends and incidence of patients with psychiatric illnesses who visited the ED during the pandemic. Studies conducted in a single hospital reported that the number of patients who visited the ED presenting with a psychiatric illness decreased during the pandemic, and the number of total ED visits during the pandemic decreased as well [21,22]. However, these findings are difficult to generalize, and the trends associated with patients who visited the ED presenting with a psychiatric illness during the pandemic have not yet been investigated.

We hypothesized that the incidence of patients with psychiatric illnesses visiting the ED would increase during the pandemic and that the trends would differ by the subgroups of psychiatric illness, such as self-harm injury, psychosis, and mood disorders. This study aimed to investigate the effect of the COVID-19 pandemic on the trends and characteristics of patients with psychiatric illnesses who visited EDs using a nationwide database.

## 2. Materials and Methods

### 2.1. Study Design and Data Sources

A cross-sectional study was conducted using the National Emergency Department Information System (NEDIS) database. The NEDIS database, an ED-based database, was created in 2013 by the Ministry of Health and Welfare and operated by the National Emergency Medical Center to monitor the quality of ED management. The database collects clinical and administrative information of patients who visit any of the 402 nationwide EDs in real time, including demographic information, prehospital information, as well as ED and hospital information. Information based on the medical records of each institution is automatically uploaded to a central government server within 14 days of the patient’s discharge from the ED or hospital. Designated and trained coordinators from each institution manage the upload process of the NEDIS data.

### 2.2. Study Setting

Korea has approximately 50 million people living in 17 administrative divisions. The Ministry of Health and Welfare has classified EDs into three levels according to capacity and resources: 38 regional EDs (Level 1), 125 local EDs (Level 2), and 239 emergency facilities (Level 3), that are operational as of 2020. There are currently no nationwide standard operating protocols in EDs for patients with psychiatric illnesses or mental health problems.

The first case of the COVID-19 pandemic in Korea was confirmed on 20 January 2020. As the number of patients increased exponentially, the Korean government raised the crisis warning level for infectious diseases to level 3 (Alert) on 27 January 2020. Once community spread of the virus had been identified on 23 February 2020, the national crisis warning level was raised to the highest level (Serious) to prevent the spread of the infection. On 29 February 2020, a nationwide social distancing strategy was implemented. Schools and workplaces were closed during that period, and many people stayed at home or were socially isolated away from home [23]. The Ministry of Health and Welfare has been operating mental health support programs for patients tested positive for COVID-19 as well as individuals in quarantine, such as screening and counseling, since February 2020. However, there have been no nationwide intervention programs for patients with psychiatric illnesses or mental health problems during this pandemic.

### 2.3. Study Population

The study population included patients who visited any of the 402 nationwide EDs during the COVID-19 pandemic (27 January to 29 June in 2020, during-COVID period) and the corresponding period 1 year prior (28 January to 30 June 2019, before-COVID period). The during-COVID period considered in this study spanned 22 weeks starting from 27 January 2020, at the initial stages of the pandemic when the national crisis warning level for infectious diseases was raised to level 3. The same period (22 weeks) corresponding to the previous year was included in the study to control for seasonal influences on the study outcomes. Patients who visited the ED for issuing a medical certificate or non-medical purposes were excluded.

### 2.4. Study Outcomes and Variables

The primary outcome of this study was the diagnosis of a psychiatric illness while visiting the ED or hospital. The NEDIS data includes multiple diagnostic codes at the ED and at discharge from the hospital, including one principal diagnosis code. The diagnosis code was collected based on the International Classification of Diseases, 10th edition (ICD-10).

For outcome measures, visits to the ED with a presentation of psychiatric illness were classified into five groups based on the diagnosis. The first group was defined as all patients with more than one psychiatric diagnosis (mental and behavioral disorders, F00–F99 of ICD-10) at the ED or during discharge from the hospital (all psychiatric disorders group). The second group included patients who visited Level 1 and Level 2 EDs for injury due to self-harm or suicidal attempts from among the first group of patients (psychiatric self-harm group). The third group included patients who visited the ED with a psychiatric diagnosis (F00–F99 of ICD-10) as the principal diagnosis (principal psychiatric group). The fourth group included patients whose principal diagnosis was a psychosis-related diagnosis (schizophrenia, schizotypal, and delusional disorders, F20–F29 of ICD-10; principal psychosis group). The fifth group included patients whose principal diagnosis was a mood disorder (mood affective disorders, F30–F39 of ICD-10; principal mood disorder group).

We collected the following information from the NEDIS database: demographics (age, sex, insurance (Medicare, Medicaid, and others)), prehospital and ED (use of emergency medical services when visiting the ED, level of ED, reason of visit (medical illness and injury), intentionality of injury, and length of stay in the ED (hours)), disposition (ED discharge diagnosis (ICD-10 code, multiple choice), hospital discharge diagnosis in case of hospitalization (ICD-10 code, multiple choice), ED disposition, and hospital disposition. Information on the intentionality of injury (self-harm) was collected only from the Level 1 and Level 2 EDs.

### 2.5. Statistical Analysis

Descriptive analysis was performed to compare the characteristics of patients in the before-COVID and during-COVID periods. The incidence rate of all ED visits and ED visits with the presentation of psychiatric illness per 1,000,000 person-days was calculated using the 2019 mid-year census population, obtained from Statistics Korea [24].

To increase the comparability, the age- and sex-standardized rates of patients with psychiatric illnesses per 100,000 patients visiting EDs were calculated using a direct standardization method which included the entire study population (total ED visits during the 44 weeks in 2019 and 2020) as a standard population. Rate differences and rate ratios of age- and sex-standardized rates are presented for the during-COVID period compared to those in the before-COVID period.

We conducted an interrupted time-series analysis to evaluate the effects of the pandemic on the rate of ED visits with the presentation of a psychiatric illness per 100,000 ED visits. Using a segmented Poisson regression model, we analyzed weekly trends of the outcomes in both periods (before-COVID and during-COVID), estimated the effect size (rate ratio (RR) over two periods; effects of the intervention) considering the underlying weekly trends, and tested the interaction effects of both periodic and weekly trends (both periods × week). We also applied seasonal models using harmonic terms that controlled for seasonal influences of time-series analyses.

Data management and statistical analyses were conducted using SAS software version 9.4 (SAS Institute Inc., Cary, NC, USA) and R statistical software (version 4.0.3; RStudio, Inc., Boston, MA, USA). The threshold for statistical significance was set at *p* < 0.05.

## 3. Results

### 3.1. Demographic Findings

#### 3.1.1. All Emergency Department (ED) Visits and Psychiatric Illness Visits

The number of overall ED visits during the entire study period (44 weeks across 2019 and 2020) were 6,210,613. Among them, 88,520 (2.5%) patients in the before-COVID period and 73,281 (2.7%) patients in the during-COVID period had a diagnosis of psychiatric illness (all psychiatric disorders group, *p*-value < 0.01), and 4240 (0.12%) patients in the before-COVID period and 4431 (0.17%) patients in the during-COVID period had visited the ED for injuries caused by self-harm with a diagnosis of psychiatric illness (psychiatric self-harm group, *p*-value < 0.01). The incidence rate of all psychiatric disorders per 1,000,000 person-days was 11.2 in the before-COVID period and 9.3 in the during-COVID period (*p*-value < 0.01), and that of self-harm with a diagnosis of psychiatric illness was 0.54 in the before-COVID period and 0.56 in the during-COVID period (*p*-value = 0.04). Among all the groups, the proportion of patients with psychiatric illness as the principal diagnosis had decreased in the during-COVID period compared to the before-COVID period (46.7% and 48.8%, respectively; *p*-value < 0.01) (Table 1). The distribution of the principal diagnostic codes of the all psychiatric disorders group is presented in the Appendix A.

#### 3.1.2. Patients with Psychiatric Diagnostic Code as Principal Diagnosis

The incidence rate of a psychiatric disorder as the principal diagnosis per 1,000,000 person-days was 5.46 in the before-COVID period and 4.33 in the during-COVID period. The incidence rate of psychosis-related diagnosis as a principal diagnosis was 0.33 in the before-COVID period and 0.30 in the during-COVID period, and that of mood disorder as the principal diagnosis was 0.64 in the before-COVID period and 0.60 in the during-COVID period (all *p*-value < 0.01) (Table 2).

### 3.2. Number of Patients with Psychiatric Illness Per 100,000 ED Visits

#### 3.2.1. Weekly Trends of Patients with Psychiatric Illnesses

The distribution and weekly trends associated with patients with psychiatric illness per 100,000 patients visiting the ED during the before-COVID and during-COVID periods are shown in Figure 1.

#### 3.2.2. Age- and Sex-Standardized Rates of Patients with Psychiatric Illness

Age- and sex-standardized rates of patients with psychiatric illness per 100,000 ED visits increased in the during-COVID period compared to the before-COVID period, except for the principal psychiatric group, as demonstrated in Table 3. Rate differences (95% CIs) of age- and sex-standardized rates were 45.7 (20.1–71.4) for all psychiatric disorders and 42.2 (36.2–48.3) for psychiatric self-harm. RRs (95% CIs) of age- and sex-standardized rates were 1.02 (1.01–1.03) for all psychiatric disorders and 1.35 (1.29–1.41) for psychiatric self-harm.

#### 3.2.3. Segmented Poisson Regression Analysis for Patients with Psychiatric Illnesses

In the segmented Poisson regression analyses, the largest estimate was observed for psychiatric self-harm (RR (95% CI), 1.53 (1.33–1.77)), followed by principal psychosis and principal mood disorder (RRs (95% CIs), 1.22 (1.05–1.42) and 1.20 (1.07–1.35)) (Table 4).

## 4. Discussion

Using a nationwide emergency patient database, this study evaluated the impact of the COVID-19 pandemic on the trends and characteristics of patients with psychiatric illnesses who visited EDs during this time. During the pandemic, the incidence rate of patients who visited the ED for injury due to self-harm or suicide attempts with a concurrent psychiatric diagnosis increased compared to that before the outbreak (incidence rate per 1,000,000 person-days: 0.54 in the before-COVID and 0.56 during-COVID periods; age- and sex-standardized rates per 100,000 ED visits: 122 and 164; proportions among all ED visits: 0.1% and 0.2%, respectively). Although the total incidence of patients with psychiatric diagnoses decreased during the pandemic, the age- and sex-standardized rates increased (incidence rate per 1,000,000 person-days: 11.2 in the before-COVID and 9.3 during-COVID periods; age- and sex-standardized rates per 100,000 ED visits: 2588 and 2634; proportions among all ED visits: 2.5% and 2.7%, respectively). During the pandemic, there was a marked increase in the cases of psychiatric self-harm, principal psychosis, and principal mood disorder (RRs (95% CIs): 1.53 (1.33 to 1.77), 1.22 (1.05 to 1.42), and 1.20 (1.07 to 1.35), respectively). These results emphasize the importance of developing effective mental health intervention programs to cope with pandemic-induced risks, as well as the need for tailored strategies to screen and support a high-risk population.

Infectious disease epidemics significantly disrupt people’s daily lives [3]. Shutdowns of workplaces and schools, coupled with mandatory social distancing affect the mental health of the public [13]. Quarantine is another deteriorating factor for mental health, and people in isolation complain of various mental health problems such as worry, nervousness, confusion, and anger [11,25]. Since infectious diseases are contagious, pandemics affect the mental health of not only individuals but also entire communities [12]. During the pandemic, 10–30% of the general population reported feeling anxious about coming in contact with the virus, and increased levels of anxiety and depression [12,16]. While the pandemic affects the mental health of all people, the most vulnerable are patients with previously diagnosed mental disorders [18]. In these cases, existing mental health conditions may be exacerbated or a new problem may also arise due to the disruption of usual care patterns during the lockdown or quarantine [5,20].

During the pandemic, the use of healthcare services for medical illnesses other than those directly associated with the virus have decreased. During the SARS epidemic in 2003, the H1N1 epidemic in 2009, and the MERS epidemic in 2015, the rate of use of healthcare services decreased primarily due to the fear of contracting the infection [26,27,28,29,30]. Even during the COVID-19 pandemic, several countries have reported decreased ED visits [3,4,31] including ED visits due to psychiatric illnesses [21,22]. However, since mental illness is highly likely to worsen during the pandemic, this requires a closer investigation. While the overall number of ED visits as well as the number of ED visits due to psychiatric illnesses decreased, the proportion of cases of self-harm and drug-overdose increased [31], and psychiatric inpatient admission remained stable during the pandemic [4]. In addition, the proportion of patients who visited the ED with a diagnosis of psychiatric illness such as psychosis and mood disorder increased during the pandemic in this study.

Psychiatric self-harm is an extreme behavior that is highly associated with suicide. In surveys conducted in the UK, more than 50% of the respondents reported psychological or physical abuse and the thought of suicide/self-harm during the lockdown period [32]. However, other studies reported that suicide did not increase but rather decreased in the general population at the early stage of the pandemic [33]. Although an increase in thoughts of suicide/self-harm in the general population did not lead to an increase in suicide, there was an increase in the self-harm ED visits among patients with underlying psychiatric illness, which may be evidence of poorly controlled mental illness during the pandemic [34,35]. People with and without previously diagnosed mental disorders became vulnerable to mental health issues during the pandemic [19,20]. However, stay-at-home regulations and the fear of transmission of the virus can be barriers to the proper use of healthcare services for patients with psychiatric illnesses as well as other populations. The decrease in the rate of ED visits due to psychiatric illnesses during the pandemic may be evidence that psychiatric patients are unable or unwilling to access healthcare services at a time when they need it the most.

The strength of this study is that we have used nationwide ED data to evaluate the incidence of ED visits of patients with psychiatric illnesses. Previous studies that have investigated the rate of ED visits of patients with psychiatric illnesses during the COVID-19 pandemic were conducted with data from a single center or several hospitals, and it is difficult to generalize these results to an entire country [4,22,31]. This is, to the best of our knowledge, the first study on the impact of COVID-19 on the incidence of ED visits of patients with psychiatric illnesses at the national level. Understanding the impact of a pandemic is the first step towards preparing for a proper response. There is a strong need to investigate the use of healthcare services by patients with psychiatric illnesses in detail and eliminate the barriers between these patients and their visits to the ED, to correctly utilize the appropriate healthcare services to treat mental health problems that are likely to worsen during the pandemic.

This study has several limitations. First, since this study is an observational study and not a randomized controlled trial, there is a limitation in causal reasoning. Our analysis was based on the assumption that there were no other meaningful factors affecting the rate of ED visits (other than COVID-19) during the study period. This assumption might not be completely true. However, interrupted time-series analysis is a quasi-experimental alternative method that compensates for the shortcomings of observational studies. Second, patients with psychiatric illnesses were defined based on the diagnosis codes in the NEDIS database. It is impossible to distinguish whether the patients with psychiatric illness who visited the ED had previously diagnosed mental disorders or were newly diagnosed during their visits to the ED. This is limited to the in-depth interpretation of the results of this study. Third, the information on the intentionality of injury (self-harm) was collected only from the Level 1 and Level 2 EDs. Level 3 ED collects only core variables. There exists the possibility of underestimation due to the limitation of the NEDIS database.

## 5. Conclusions

During the COVID-19 pandemic, the incidence of patients visiting the ED for self-harm or suicide attempts with a concurrent diagnosis of psychiatric illness increased; however, the incidence of visits by those with any general psychiatric diagnosis decreased. By subgroup, the proportion of patients with psychosis and mood disorder as the principal diagnosis who visited the ED also increased during the pandemic. Developing effective mental health intervention programs and tailored strategies for screening and supporting high-risk populations are needed to cope with pandemic-induced risks for mental health.

## Figures and Tables

**Figure 1 jcm-11-00488-f001:**
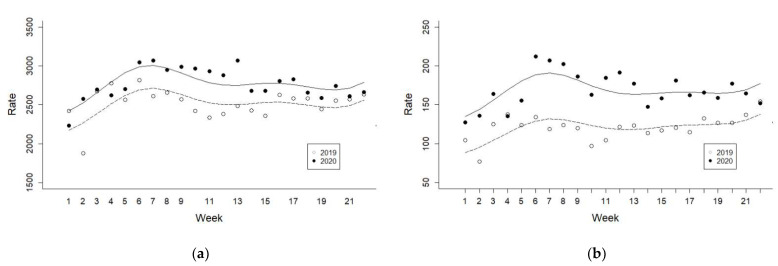
Weekly trends of patients with psychiatric illnesses per 100,000 patients visiting the ED during and before the COVID-19 pandemic. All y axes are the rate of patients per 100,000 ED visits (**a**) all psychiatric disorders, (**b**) psychiatric self-harm, (**c**) principal psychiatric, (**d**) principal psychosis, and (**e**) principal mood disorder.

**Table 1 jcm-11-00488-t001:** Demographics of all patients and patients with psychiatric illness who visited the emergency department (ED).

	Total ED Visits				All Psychiatric Disorders				
	Before-COVID		During-COVID		Before-COVID		During-COVID		*p*-Value
	N	%	N	%	N	%	N	%	
Total	3,540,628		2,669,985		88,520		73,281		
Incidence rate, per 1,000,000 person-days *	447.8		337.7		11.2		9.3		<0.01
Age, year									<0.01
0~18	752,500	21.3	364,372	13.6	4394	5.0	3047	4.2	
10~64	1,993,876	56.3	1,628,517	61.0	52,196	59.0	41,686	56.9	
65~120	794,252	22.4	677,096	25.4	31,930	36.1	28,548	39.0	
Median (IQR)	45 (22–63)		48 (28–65)		56 (39–75)		57 (39–77)		<0.01
Sex, female	1,735,274	49.0	1,281,076	48.0	47,164	53.3	39,286	53.6	0.19
EMS use	595,780	16.8	521,961	19.5	37,685	42.6	31,955	43.6	<0.01
Level of ED, 1 and 2	2,467,934	69.7	1,827,591	68.4	63,798	72.1	53,396	72.9	<0.01
Reason of visit									<0.01
Medical illness	2,557,421	72.2	1,882,434	70.5	73,129	82.6	59,352	81.0	
Injury	983,207	27.8	787,551	29.5	15,391	17.4	13,929	19.0	
Self-harm injury	14,797	0.4	14,304	0.5	4240	4.8	4431	6.0	<0.01
Incidence rate, per 1,000,000 person-days *	1.87		1.81		0.54	0.56		0.04	
Clinical outcomes									
Admission	674,417	19.0	556,363	20.8	36,902	41.7	33,521	45.7	<0.01
In-hospital mortality	46,572	1.3	46,352	1.7	1954	2.2	1902	2.6	<0.01
ED	17,702	0.5	18,696	0.7	121	0.1	123	0.2	
Ward	28,870	0.8	27,656	1.0	1833	2.1	1779	2.4	
ED visit with psychiatric illness									
All psychiatric disorders	88,520	2.5	73,281	2.7	88,520	100.0	73,281	100.0	
Psychiatric self-harm	4240	0.1	4431	0.2	4240	4.8	4431	6.0	<0.01
Principal psychiatric	43,171	1.2	34,249	1.3	43,171	48.8	34,249	46.7	<0.01
Principal psychosis	2716	0.1	2424	0.1	2716	3.1	2424	3.3	0.01
Principal mood disorder	5191	0.1	4839	0.2	5191	5.9	4839	6.6	<0.01

* Incidence rate per 100,000 person-days was calculated using the 2019 mid-year Census population. COVID-19, coronavirus disease 2019; ED, emergency department; IQR, interquartile range; EMS, emergency medical services.

**Table 2 jcm-11-00488-t002:** Demographics of all patients and patients with psychiatric illness who visited the ED.

	Principal Psychiatric Group				*p*-Value
	Before-COVID		During-COVID		
	N	%	N	%	
Total	43,171		34,249		
Incidence rate, per 1,000,000 person-days *	5.46		4.33		<0.01
Age, year					<0.01
0~18	3230	7.5	2220	6.5	
10~64	31,430	72.8	25,179	73.5	
65~120	8511	19.7	6850	20.0	
Median (IQR)	48 (30–61)		47 (29–61)		0.58
Sex, female	23,614	54.7	18,996	55.5	0.03
Insurance, Medicaid	6455	15.0	5163	15.1	0.63
EMS use	18,786	43.5	14,589	42.6	0.01
Level of ED, 1 and 2	30,087	69.7	24,217	70.7	<0.01
Reason of visit, injury	6069	14.1	5146	15.0	<0.01
Length of stay in the ED, Median (IQR), hours	2.4 (1.4–4.3)		2.3 (1.3–4.5)		0.67
Clinical outcomes					
Admission	7140	16.5	5968	17.4	<0.01
In-hospital mortality	177	0.4	181	0.5	0.02
ED	43	0.1	32	0.1	
Ward	134	0.3	149	0.4	
Diagnosis (ICD-10)					
Schizophrenia and so on (F2)	2625	6.1	2346	6.8	0.01
Incidence rate, per 1,000,000 person-days *	0.33		0.30		<0.01
Schizophrenia (F20)	1707	4.0	1519	4.4	
Schizotypal disorder (F21)	1	0.0	8	0.0	
Persistent delusional disorders (F22)	86	0.2	51	0.1	
Acute and transient psychotic disorders (F23)	192	0.4	160	0.5	
Induced delusional disorder (F24)	5	0.0	1	0.0	
Schizoaffective disorders (F25)	215	0.5	234	0.7	
Other nonorganic psychotic disorders (F28)	30	0.1	21	0.1	
Unspecified nonorganic psychosis (F29)	389	0.9	352	1.0	
Mood disorders (F3)	5068	11.7	4726	13.8	<0.01
Incidence rate, per 1,000,000 person-days *	0.64		0.60		<0.01
Manic episode (F30)	42	0.1	29	0.1	
Bipolar affective disorder (F31)	1744	4.0	1538	4.5	
Depressive episode (F32)	2745	6.4	2248	6.6	
Recurrent depressive disorder (F33)	326	0.8	355	1.0	
Persistent mood disorders (F34)	91	0.2	302	0.9	
Other mood disorders (F38)	8	0.0	19	0.1	
Unspecified mood disorder (F39)	112	0.3	235	0.7	
Others	35,478	82.2	27,177	79.4	

* Incidence rate per 100,000 person-days was calculated using the 2019 mid-year Census population. COVID-19, coronavirus disease 2019; ED, emergency department; IQR, interquartile range; EMS, emergency medical services; ICD-10, International Classification of Diseases, 10th edition.

**Table 3 jcm-11-00488-t003:** Age- and sex-standardized rates of patients with psychiatric illness per 100,000 patients visiting EDs.

	Before-COVID			During-COVID			Rate Difference			Rate Ratio		
	Rate *	95% CI		Rate *	95% CI		Diff	95% CI		Ratio	95% CI	
All psychiatric disorders	2588	2571	2605	2634	2615	2653	45.7	20.1	71.4	1.02	1.01	1.03
Psychiatric self-harm	122	118	125	164	159	169	42.2	36.2	48.3	1.35	1.29	1.41
Principal psychiatric	1251	1239	1262	1248	1235	1261	−2.6	−20.4	15.1	1.00	0.98	1.01
Principal psychosis	76	73	79	85	82	88	8.9	4.4	13.4	1.12	1.06	1.18
Principal mood disorder	145	141	149	176	171	181	31.1	24.7	37.5	1.21	1.17	1.26

* Age- and sex-standardized rates of ED visits with psychiatric illnesses per 100,000 ED visits were calculated using total ED visits during 44 weeks in 2019 and 2020 as a standard population. CI, confidence interval; COVID-19, coronavirus disease 2019; ED, emergency department.

**Table 4 jcm-11-00488-t004:** Segmented Poisson regression analysis for patients with psychiatric illnesses per 100,000 patients visiting EDs during the 22 weeks in 2020 compared to same period in the previous year.

	Rate Ratio *	95% CI	
All psychiatric disorders	1.12	1.03	1.20
Psychiatric self-harm	1.53	1.33	1.77
Principal psychiatric	1.04	0.97	1.11
Principal psychosis	1.22	1.05	1.42
Principal mood disorder	1.20	1.07	1.35

* Rate ratios of ED visits with psychiatric illnesses per 100,000 ED visits were adjusted for week, interaction term (both periods × week), and harmonic term. CI, confidence interval; ED, emergency department.

## Data Availability

The data of this study were obtained from the National Emergency Medical Center under the Ministry of Health and Welfare in Korea but restrictions apply to the availability of these data and so are not publicly available.

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
