# Peer review of "Impact of the COVID-19 Pandemic on the Incidence and Characteristics of Patients with Psychiatric Illnesses Visiting Emergency Departments in Korea"

_jcm, 2022, doi:10.3390/jcm11030488_

Round 1

Reviewer 1 Report

Dear Authors,

Manuscript reports on a well-written, sound, robust study, based on comprehensive national (Korean) (observational) data used in order to estimate emergency department (ED) use by patients with psychiatric illnesses.  

A study is well designed (e.g. appropriate use of periods for interrupted time-series analysis), well-powered (e.g. around 6 000 000 cases of “overall ED”, while around 150 000 “all psychiatric ED” uses), statistical analyses preformed are adequate (especially Segmented Poisson regression). Results are clearly presented, except all figures (in manuscript under Figure 1.) which are of very low resolution which makes them unusable. Limitations are clearly stated. Conclusions from the results of the study. All in all, according to my view this study adds significant and timely value to very important topic – estimation of the influence of COVID-19 context on mental health and especially persons with mental health disorders.

A possibilities of improvement include:

  1. Broadening introduction section especially in terms of using more recent references.

In that sense, if I might suggest:

Alongside references [5,6] on Page 2. (and later on) references:

Kuzman, M. R., Curkovic, M., & Wasserman, D. (2020). Principles of mental health care during the COVID-19 pandemic. European psychiatry : the journal of the Association of European Psychiatrists63(1), e45. https://doi.org/10.1192/j.eurpsy.2020.54

COVID-19 Mental Disorders Collaborators (2021). Global prevalence and burden of depressive and anxiety disorders in 204 countries and territories in 2020 due to the COVID-19 pandemic. Lancet (London, England)398(10312), 1700–1712. https://doi.org/10.1016/S0140-6736(21)02143-7

Alongside references [7] on Page 2. (and later on) reference:

Bourmistrova, N. W., Solomon, T., Braude, P., Strawbridge, R., & Carter, B. (2021). Long-term effects of COVID-19 on mental health: A systematic review. Journal of affective disorders299, 118–125. Advance online publication. https://doi.org/10.1016/j.jad.2021.11.031

Alongside references [9-11] on Page 2. (and later on) reference:

Manchia, M., Gathier, A. W., Yapici-Eser, H., Schmidt, M. V., de Quervain, D., van Amelsvoort, T., Bisson, J. I., Cryan, J. F., Howes, O. D., Pinto, L., van der Wee, N. J., Domschke, K., Branchi, I., & Vinkers, C. H. (2021). The impact of the prolonged COVID-19 pandemic on stress resilience and mental health: A critical review across waves. European neuropsychopharmacology : the journal of the European College of Neuropsychopharmacology55, 22–83. Advance online publication. https://doi.org/10.1016/j.euroneuro.2021.10.864

Alongside reference [13] on Page 2. (and later on) reference:

Hassan, L., Peek, N., Lovell, K., Carvalho, A. F., Solmi, M., Stubbs, B., & Firth, J. (2021). Disparities in COVID-19 infection, hospitalisation and death in people with schizophrenia, bipolar disorder, and major depressive disorder: a cohort study of the UK Biobank. Molecular psychiatry, 10.1038/s41380-021-01344-2. Advance online publication. https://doi.org/10.1038/s41380-021-01344-2

Alongside reference [9] on the first page in discussion section reference:

Ćurković, M., Košec, A., & Ćurković, D. (2020). Math and aftermath of COVID-19 pandemic and its interrelationship from the resilience perspective. The Journal of infection81(2), e173–e174. https://doi.org/10.1016/j.jinf.2020.06.020

  1. In discussion section:

I would like to see discussion of study’s findings – especially those speaking of raising incidence of ED patients because of self-harm, suicide attempts with underlying psychiatric illness with those from:

Pirkis, J., John, A., Shin, S., DelPozo-Banos, M., Arya, V., Analuisa-Aguilar, P., Appleby, L., Arensman, E., Bantjes, J., Baran, A., Bertolote, J. M., Borges, G., Brečić, P., Caine, E., Castelpietra, G., Chang, S. S., Colchester, D., Crompton, D., Curkovic, M., Deisenhammer, E. A., … Spittal, M. J. (2021). Suicide trends in the early months of the COVID-19 pandemic: an interrupted time-series analysis of preliminary data from 21 countries. The lancet. Psychiatry8(7), 579–588. https://doi.org/10.1016/S2215-0366(21)00091-2

Or other similar studies which are speaking of decreased general/publics’ rates of suicides in similar time periods while using somewhat similar design and statistical analysis (it belongs somewhere were references [27, 28] first appear in the manuscript).

  1. Above mentioned figures are of inadequate quality – resolution, at least in current version of the manuscript.
  2. The reasons for not using data on self-harm and suicides from Level 3 ED’s (a seemingly significant proportion of the ED system), although stressed in limitations section, is not elaborated. I would like to see why these data was not used as this is one of the principal findings from the study (although I might assume that cause is a some kind of technical detail).  
  3. In the first page of Discussion section sentence I suggest remodeling sentence:

“In these cases, the existing mental health conditions may be exacerbated or a new problem may ALSO arise due to the lack of appropriate treatment during the lockdown or quarantine.”

Instead of “lack of appropriate” I would use phrases as “disruption of usual patterns” or something like that.

  1. In the same page insert word into sentence flowingly:

During the SARS epidemic in 2003, the H1N1 epidemic in 2009, and the MERS epidemic in 2015, the rate of use of healthcare services decreased PRIMARALY due to the fear of contracting the infection.[21-25]

  1. On the next page correct sentence:

The decrease in the rate of ED visits due to psychiatric illnesses during the pandemic may be evidence that psychiatric patients are unable OR UNWILLING to access healthcare services at a time when they need it the most.

  1. I would also like to see your very brief interpretation of, at a first sight conflicting results, on the same variables when using proportions, rates and standardizes rates (might belong to discussion or even limitation section).
  2. Additionally, I would like to see reasons behind omitting certain psychiatric diagnostic groups, that is, anxious disorders diagnoses (and to a certain extent substance abuse – F1x) – these are the clusters of disorders which increase can also be expected within COVID-19 context.
  3. Also, how you handled the “missing data” – I am actually not sure if this is an actual issue as inputs to database are probably mandatory.

Once again, applause for the great work and great study which contribution is certainly significant.

Author Response

Reviewer 1

Dear Editor, dear Authors,

First of all, thank you for the opportunity to review given manuscript.

Manuscript reports on a well-written, sound, robust study, based on comprehensive national (Korean) (observational) data used in order to estimate emergency department (ED) use by patients with psychiatric illnesses. 

A study is well designed (e.g. appropriate use of periods for interrupted time-series analysis), well-powered (e.g. around 6 000 000 cases of “overall ED”, while around 150 000 “all psychiatric ED” uses), statistical analyses preformed are adequate (especially Segmented Poisson regression). Results are clearly presented, except all figures (in manuscript under Figure 1.) which are of very low resolution which makes them unusable. Limitations are clearly stated. Conclusions from the results of the study. All in all, according to my view this study adds significant and timely value to very important topic – estimation of the influence of COVID-19 context on mental health and especially persons with mental health disorders.

(ANSWER) Thank you for the review and the valuable comments. Each comment is addressed as you pointed.

A possibilities of improvement include:

  1. Broadening introduction section especially in terms of using more recent references.

In that sense, if I might suggest:

Alongside references [5,6] on Page 2. (and later on) references:

Kuzman, M. R., Curkovic, M., & Wasserman, D. (2020). Principles of mental health care during the COVID-19 pandemic. European psychiatry : the journal of the Association of European Psychiatrists, 63(1), e45. https://doi.org/10.1192/j.eurpsy.2020.54

COVID-19 Mental Disorders Collaborators (2021). Global prevalence and burden of depressive and anxiety disorders in 204 countries and territories in 2020 due to the COVID-19 pandemic. Lancet (London, England), 398(10312), 1700–1712. https://doi.org/10.1016/S0140-6736(21)02143-7

Alongside references [7] on Page 2. (and later on) reference:

Bourmistrova, N. W., Solomon, T., Braude, P., Strawbridge, R., & Carter, B. (2021). Long-term effects of COVID-19 on mental health: A systematic review. Journal of affective disorders, 299, 118–125. Advance online publication. https://doi.org/10.1016/j.jad.2021.11.031

Alongside references [9-11] on Page 2. (and later on) reference:

Manchia, M., Gathier, A. W., Yapici-Eser, H., Schmidt, M. V., de Quervain, D., van Amelsvoort, T., Bisson, J. I., Cryan, J. F., Howes, O. D., Pinto, L., van der Wee, N. J., Domschke, K., Branchi, I., & Vinkers, C. H. (2021). The impact of the prolonged COVID-19 pandemic on stress resilience and mental health: A critical review across waves. European neuropsychopharmacology : the journal of the European College of Neuropsychopharmacology, 55, 22–83. Advance online publication. https://doi.org/10.1016/j.euroneuro.2021.10.864

Alongside reference [13] on Page 2. (and later on) reference:

Hassan, L., Peek, N., Lovell, K., Carvalho, A. F., Solmi, M., Stubbs, B., & Firth, J. (2021). Disparities in COVID-19 infection, hospitalisation and death in people with schizophrenia, bipolar disorder, and major depressive disorder: a cohort study of the UK Biobank. Molecular psychiatry, 10.1038/s41380-021-01344-2. Advance online publication. https://doi.org/10.1038/s41380-021-01344-2

Alongside reference [9] on the first page in discussion section reference:

Ćurković, M., Košec, A., & Ćurković, D. (2020). Math and aftermath of COVID-19 pandemic and its interrelationship from the resilience perspective. The Journal of infection, 81(2), e173–e174. https://doi.org/10.1016/j.jinf.2020.06.020

(ANSWER) Thank you for the review. I added all the references you recommended in the Introduction section.

(REVISION: Introduction and References)

  1. In discussion section:

I would like to see discussion of study’s findings – especially those speaking of raising incidence of ED patients because of self-harm, suicide attempts with underlying psychiatric illness with those from:

Pirkis, J., John, A., Shin, S., DelPozo-Banos, M., Arya, V., Analuisa-Aguilar, P., Appleby, L., Arensman, E., Bantjes, J., Baran, A., Bertolote, J. M., Borges, G., Brečić, P., Caine, E., Castelpietra, G., Chang, S. S., Colchester, D., Crompton, D., Curkovic, M., Deisenhammer, E. A., … Spittal, M. J. (2021). Suicide trends in the early months of the COVID-19 pandemic: an interrupted time-series analysis of preliminary data from 21 countries. The lancet. Psychiatry, 8(7), 579–588. https://doi.org/10.1016/S2215-0366(21)00091-2

Or other similar studies which are speaking of decreased general/publics’ rates of suicides in similar time periods while using somewhat similar design and statistical analysis (it belongs somewhere were references [27, 28] first appear in the manuscript).

(ANSWER) Thank you for the review. I added more reviews in the Discussion section and added all the references you recommended accordingly.

(REVISION: Discussion and References)

Psychiatric self-harm is an extreme behavior that is highly associated with suicide. In surveys conducted in the UK, more than 50% of the respondents reported psychological or physical abuse and the thought of suicide/self-harm during the lockdown period.[32] However, other studies reported that suicide did not increase but rather decreased in the general population at the early stage of the pandemic.[33] Although an increase in thoughts of suicide/self-harm in the general population did not lead to an increase in suicide, there was an increase in the self-harm ED visits among patients with underlying psychiatric illness, which may be evidence of poorly controlled mental illness during the pandemic.[34,35] People with and without previously diagnosed mental disorders become vulnerable to mental health issues during the pandemic.[19,20] However, stay-at-home regulations and the fear of transmission of the virus can be barriers to the proper use of healthcare services for patients with psychiatric illnesses as well as other populations. The decrease in the rate of ED visits due to psychiatric illnesses during the pandemic may be evidence that psychiatric patients are unable or unwilling to access healthcare services at a time when they need it the most.

  1. Above mentioned figures are of inadequate quality – resolution, at least in current version of the manuscript.

(ANSWER) Thank you for the review. I modified the Figure 1 accordingly.

(REVISION: Figure 1)

  1. The reasons for not using data on self-harm and suicides from Level 3 ED’s (a seemingly significant proportion of the ED system), although stressed in limitations section, is not elaborated. I would like to see why these data was not used as this is one of the principal findings from the study (although I might assume that cause is a some kind of technical detail).

(ANSWER) Thank you for the review. The Ministry of Health and Welfare has classified EDs into three levels according to capacity and resources: 38 regional EDs (Level 1), 125 local EDs (Level 2), and 239 emergency facilities (Level 3), that are operational as of 2020. Level 3 EDs have the most in number but the lowest capacity. Level 1 and 2 EDs required input of all variables in the NEDIS database, but some variables were not mandatory variables for Level 3 EDs. The intentionality of injury is not a core variable in NEDIS data and therefore was not collected in the Level 3 EDs. I revised the paragraph accordingly.

(REVISION: Limitation)

Third, the information on the intentionality of injury (self-harm) was collected only from the Level 1 and Level 2 EDs. Level 3 ED collects only core variables. There exists the possibility of underestimation due to the limitation of the NEDIS database.

  1. In the first page of Discussion section sentence I suggest remodeling sentence:

“In these cases, the existing mental health conditions may be exacerbated or a new problem may ALSO arise due to the lack of appropriate treatment during the lockdown or quarantine.”

Instead of “lack of appropriate” I would use phrases as “disruption of usual patterns” or something like that.

(ANSWER) Thank you for the review. I revised the sentence accordingly.

(REVISION: Discussion)

In these cases, the existing mental health conditions may be exacerbated or a new problem may also arise due to the disruption in usual care patterns during the lockdown or quarantine.

  1. In the same page insert word into sentence flowingly:

During the SARS epidemic in 2003, the H1N1 epidemic in 2009, and the MERS epidemic in 2015, the rate of use of healthcare services decreased PRIMARALY due to the fear of contracting the infection.[21-25]

(ANSWER) Thank you for the review. I revised the sentence accordingly.

(REVISION: Discussion)

During the SARS epidemic in 2003, the H1N1 epidemic in 2009, and the MERS epidemic in 2015, the rate of use of healthcare services decreased primarily due to the fear of contracting the infection

  1. On the next page correct sentence:

The decrease in the rate of ED visits due to psychiatric illnesses during the pandemic may be evidence that psychiatric patients are unable OR UNWILLING to access healthcare services at a time when they need it the most.

(ANSWER) Thank you for the review. I revised the sentence accordingly.

(REVISION: Discussion)

The decrease in the rate of ED visits due to psychiatric illnesses during the pandemic may be evidence that psychiatric patients are unable or unwilling to access healthcare services at a time when they need it the most.

  1. I would also like to see your very brief interpretation of, at a first sight conflicting results, on the same variables when using proportions, rates and standardizes rates (might belong to discussion or even limitation section).

(ANSWER) Thank you for the review.

(REVISION: Discussion)

During the pandemic, the incidence rate of patients who visited the ED for injury due to self-harm or suicide attempts with a concurrent psychiatric diagnosis increased compared to that before the outbreak (incidence rate per 1,000,000 person-days: 0.54 in the before-COVID and 0.56 during-COVID periods; age- and sex-standardized rates per 100,000 ED visits: 122 and 164; proportions among all ED visits: 0.1% and 0.2%, respectively). Although the total incidence of patients with psychiatric diagnoses decreased during the pandemic, the age- and sex-standardized rates increased (incidence rate per 1,000,000 person-days: 11.2 in the before-COVID and 9.3 during-COVID periods; age- and sex-standardized rates per 100,000 ED visits: 2,588 and 2,634; proportions among all ED visits: 2.5% and 2.7%, respectively).

  1. Additionally, I would like to see reasons behind omitting certain psychiatric diagnostic groups, that is, anxious disorders diagnoses (and to a certain extent substance abuse – F1x) – these are the clusters of disorders which increase can also be expected within COVID-19 context.

(ANSWER) Thank you for the review. The “all psychiatric disorders group” and “principal psychiatric group” includes all F diagnosis (mental and behavioral disorders, F00-F99 of ICD-10), including anxious disorder and substance abuse. For the subgroup, the number of patients with those diagnosis was not as large as the number of principal psychosis and the principal mood disorder groups.

  1. Also, how you handled the “missing data” – I am actually not sure if this is an actual issue as inputs to database are probably mandatory.

(ANSWER) Thank you for the review. The NEDIS data used in this study is a database operated by the Ministry of Health and Welfare and the National Emergency Medical Center to monitor the quality of ED management, and there is almost no missing data because patient data can be registered only when core variables are input. However, as some variables were not mandatory core variables for Level 3 ED (whereas Level 1 and 2 EDs should input all variables in the NEDIS database), the intentionality of injury was not collected in the Level 3 EDs. The variables used in the study were core variables, and there were no missing data.

Once again, applause for the great work and great study which contribution is certainly significant.

(ANSWER) Thank you for the review and the valuable comments.

Reviewer 2 Report

The major strength of the study is the scope that include all national emergencies data for 22 weeks in 2019 and 2022.

Several studies of suicidal behaviors during the Covid pandemia are missing in the introduction and in the references mentioned above. Some of these studies shows discondants results.

They should be included in the review an discussion of the results.

Author Response

The major strength of the study is the scope that include all national emergencies data for 22 weeks in 2019 and 2022.

(ANSWER) Thank you for the review and the valuable comments. Each comment is addressed as you pointed.

Several studies of suicidal behaviors during the Covid pandemia are missing in the introduction and in the references mentioned above. Some of these studies shows discondants results.

They should be included in the review an discussion of the results.

(ANSWER) Thank you for the review. I added more reviews of this study’s findings in the Discussion section, and added all the references recommended by reviewers in the Introduction and Discussion sections.

(REVISION: Introduction, Discussion and References)

Psychiatric self-harm is an extreme behavior that is highly associated with suicide. In surveys conducted in the UK, more than 50% of the respondents reported psychological or physical abuse and the thought of suicide/self-harm during the lockdown period.[32] However, other studies reported that suicide did not increase but rather decreased in the general population at the early stage of the pandemic.[33] Although an increase in thoughts of suicide/self-harm in the general population did not lead to an increase in suicide, there was an increase in the self-harm ED visits among patients with underlying psychiatric illness, which may be evidence of poorly controlled mental illness during the pandemic.[34,35] People with and without previously diagnosed mental disorders become vulnerable to mental health issues during the pandemic.[19,20] However, stay-at-home regulations and the fear of transmission of the virus can be barriers to the proper use of healthcare services for patients with psychiatric illnesses as well as other populations. The decrease in the rate of ED visits due to psychiatric illnesses during the pandemic may be evidence that psychiatric patients are unable or unwilling to access healthcare services at a time when they need it the most.

Round 2

Reviewer 2 Report

Although there are several other publications, the authors have included some significant ones